# Anti-Melanogenic Effects of *Cnidium monnieri* Extract via p38 Signaling-Mediated Proteasomal Degradation of Tyrosinase

**DOI:** 10.3390/plants13101305

**Published:** 2024-05-09

**Authors:** Soon Ho Choi, Hyunggun Kim, Jeon Hwang-Bo, Kyoung Mi Kim, Jeong Eun Kwon, Sung Ryul Lee, Sun Ha Hwang, Se Chan Kang, Yeong-Geun Lee

**Affiliations:** 1Research Institute, APRG Inc., Yongin 16950, Republic of Korea; shchoi@aprg.co.kr; 2Department of Biomechatronic Engineering, Sungkyunkwan University, Suwon 16419, Republic of Korea; hkim.bme@skku.edu; 3Department of Biopharmaceutical Biotechnology and Graduate School of Biotechnology, Kyung Hee University, Yongin 17104, Republic of Korea; hbj3286@khu.ac.kr (J.H.-B.); jjung@nmr.kr (J.E.K.); king-sun89@daum.net (S.H.H.); 4Research Center, CureBio Therapeutics Co., Ltd., Suwon 16229, Republic of Korea; kmkim@curebiotx.com; 5Department of Convergence Biomedical Science, Cardiovascular and Metabolic Disease Center, College of Medicine, Inje University, Busan 47392, Republic of Korea; lsr1113@inje.ac.kr

**Keywords:** melanogenesis, microphthalmia-associated transcription factor (MITF), p38 activation, α-melanocyte-stimulating hormone (α-MSH)

## Abstract

*Cnidium monnieri* fructus is widely used in traditional Oriental medicine for treating female genital disorders, male impotence, frigidity, and skin-related conditions in East Asia. However, the role of *C. monnieri* fructus extract (CMFE) in melanin synthesis is not well elucidated. This study aimed to investigate the anti-melanogenesis effect and mechanism of action of CMFE in α-MSH-stimulated B16F10 cells. Intracellular melanin content and tyrosinase activity were measured in α-MSH-stimulated B16F10 cells treated with various concentrations of CMFE (0.5–5 μg/mL). mRNA and protein levels of tyrosinase and MITF were evaluated using qRT-PCR and ting. CMFE’s effect on the proteasomal degradation of tyrosinase was confirmed using a proteasomal degradation inhibitor, MG132. CMFE treatment activated p38, a protein associated with proteasomal degradation. Treatment with CMFE at up to 5 μg/mL showed no significant cytotoxicity. CMFE significantly reduced α-MSH-stimulated melanin production (43.29 ± 3.55% decrease, *p* < 0.05) and cellular tyrosinase activity (31.14 ± 3.15% decrease, *p* < 0.05). Although mRNA levels of MITF and tyrosinase increased, CMFE suppressed tyrosinase protein levels. The suppressive effect of CMFE on tyrosinase protein was blocked by MG132. CMFE inhibited melanogenesis by promoting the proteasome degradation of tyrosinase through p38 activation. These findings suggest that CMFE has the potential to be a natural whitening agent for inhibiting melanogenesis.

## 1. Introduction

Melanogenesis is a pivotal process that is crucial to the synthesis of melanin, the pigment responsible for determining the color of the skin, hair, and eyes. The molecular structure of melanin enables it to absorb ultraviolet and visible light, effectively shielding the skin from harmful ultraviolet rays [1,2,3]. However, when melanin is produced in excess, pigment builds up in the skin, forming spots and freckles, and these lesions can lead to skin cancer. Various transcription factors and enzymes regulate melanin biosynthesis. Microphthalmia-associated transcription factor (MITF) plays a crucial role in the transcription of melanogenesis enzymes [4,5].

Various signaling pathways regulate MITF, and the cyclic adenosine monophosphate (cAMP)–protein kinase A (PKA)–cAMP responsive element binding (CREB) protein–CREB binding protein (CBP) mechanism is one of the primary pathways that upregulate the expression of MITF [6]. Additionally, the mitogen-activated protein kinase (MAPK) pathway, including extracellular signal-regulated kinase (ERK), c-Jun NH2-terminal kinase (JNK), and p38, can enhance melanogenesis by promoting MITF expression through phosphorylation [6,7]. Among the enzymes transcribed by MITF, tyrosinase catalyzes the initial two reactions of melanin synthesis: the hydroxylation of L-tyrosine to L-3,4-dihydroxyphenylalanine (L-DOPA) and the oxidation of L-DOPA to L-dopaquinone. The final steps of melanin formation involve the catalytic oxidation process that is primarily carried out by tyrosinase-related protein (TRP)-1 and TRP-2 [8,9]. Tyrosinase activity inhibitors are commonly used to treat skin conditions related to hyperpigmentation or for skin whitening. Strategies to inhibit tyrosinase activity encompass the following: (1) suppressing tyrosinase mRNA transcription, (2) disrupting tyrosinase glycosylation (maturation), (3) inhibiting tyrosinase’s catalytic activity, and (4) promoting accelerated tyrosinase degradation [5].

*Cnidium monnieri* (L.) Cuss., an annual plant of the Apiaceae family, is indigenous to regions including Korea, China, Japan, and Vietnam. It has a rich history that is deeply rooted in traditional Oriental medicine, spanning millennia [10]. *C. monnieri*, especially its fruit, has garnered scientific interest due to its notable efficacy in addressing various dermatological concerns, such as irritation, pruritus, and allergic reactions [10,11]. Its efficacy has led to the meticulous isolation and identification of several bioactive compounds, including coumarin, chromone, benzofuran, and monoterpenoids [12]. Sophisticated in vivo and in vitro investigations have revealed the remarkable pharmacological versatility of *C. monnieri* fructus, demonstrating its effectiveness as a potent agent against inflammation, allergies, itching, fungal infections, vasodilation, and tumors, as showcased in recent studies [13,14]. However, amidst this breadth of knowledge, the elusive potential of *C. monnieri* fructus to modulate melanogenesis remains an intriguing untapped area within the scientific realm.

Responding to the increasing demand for safe and naturally derived cosmetic ingredients with effective depigmentation properties, there has been a surge in rigorous scientific investigations. Our study, rooted in this evolving landscape, aimed to explore the potential of *C. monnieri* fructus extract (CMFE) in reducing melanin production. Using B16F10 mouse melanoma cells as an in vitro model, we carefully examined the mechanisms by which CMFE modulates melanogenesis. Through our meticulous exploration, we gained valuable insights into how CMFE effectively regulates melanin production, offering promising applications in cosmetic science.

## 2. Results

### 2.1. Effects of CMFE on Melanin Production in α-MSH-Stimulated B16F10 Cells

B16F10 cells were treated with various concentrations (0, 0.5, 1, 5, 10, and 20 µg/mL) of CMFE for 48 h to examine the effects on the viability of the B16F10 cells (Figure 1A). CMFE did not show any significant cell cytotoxicity at 0.5 (102.06 ± 7.90%), 1 (98.15 ± 4.31%), and 5 µg/mL (96.97 ± 7.06%) compared to untreated controls, but significant cell cytotoxicity was shown at and above 10 µg/mL (*p* < 0.05). Therefore, the maximal dose of CMFE was set at 5 µg/mL in subsequent experiments.

The effects of CMFE on melanin production and the activity of tyrosinase, a major key enzyme for melanogenesis in α-MSH-stimulated B16F10 cells, were investigated. Arbutin was used as a positive control for the determination of melanin production. As shown in Figure 1B, melanin production was significantly increased (355.13 ± 30.45% vs. untreated control, *p* < 0.05) by α-MSH, and the increased melanin production was decreased in a dose-dependent manner. The inhibitory effect of 5 μg/mL CMFE was stronger than that of the positive control with 1 mM arbutin (34.95 ± 4.15% vs. 43.29 ± 3.55%). These results indicate that CMFE treatment is effective in inhibiting α-MSH-mediated melanogenesis.

### 2.2. Effects of CMFE on Both Cellular and Mushroom Tyrosinase Activity

Tyrosinase is a major enzyme responsible for the initial pathway of melanogenesis and has been widely used as an enzyme for screening potential melanogenesis inhibitors. Cellular tyrosinase activity in α-MSH-stimulated B16F10 cells was increased by 335.88 ± 24.93% compared to the untreated control group. To confirm this hypothesis, we investigated the effect of CMFE on mushroom tyrosinase activity in a cell-free system. As shown in Figure 2A, CMFE did not inhibit the activity of mushroom tyrosinase, whereas L-ascorbic acid inhibited the activity of mushroom tyrosinase by 72.80 ± 0.13% (*p* < 0.05). These results show that intracellular tyrosinase expression is suppressed by CMFE, reducing the activity of tyrosinase and suppressing melanin production. The cellular tyrosinase activity that was increased by α-MSH was decreased in a dose-dependent manner upon CMFE treatment and was statistically significant in the 5 μg/mL CMFE treatment group (31.14 ± 3.15% inhibition, *p* < 0.05) (Figure 2B). These results indicate that the suppression of melanin production by CMFE is associated with a decrease in cellular tyrosinase activity. The reduced cellular tyrosinase activity is assumed to be due to direct inhibition of tyrosinase activity or the inhibition of intracellular tyrosinase expression.

### 2.3. Effects of CMFE on mRNA and Protein Levels of MITF and Tyrosinase

Since CMFE was shown to inhibit melanin production through the inhibition of tyrosine expression, the effects of CMFE on mRNA and protein levels of tyrosinase and MITF, a transcription factor regulating the expression of tyrosinase, were confirmed. The mRNA expressions of MITF and tyrosinase in B16F10 cells stimulated with α-MSH were significantly increased compared to the untreated control group. CMFE did not inhibit the increased mRNA expression caused by α-MSH but rather, it was further increased (Figure 3A). These results exclude the possibility that the inhibitory effects of CMFE on melanin production and cellular tyrosinase activity may be associated with decreased mRNA levels of MITF and tyrosinase. The protein levels of MITF and tyrosinase in B16F10 cells stimulated with α-MSH were significantly increased compared to the untreated control. The increased protein expression was greatly reduced by CMFE (Figure 3B,C). These results show that the decrease in tyrosinase activity caused by CMFE is related to the decrease in intracellular tyrosinase protein levels.

### 2.4. Effects of CMFE on Proteasomal Degradation of Tyrosinase Protein

CMFE suppressed the activities of cellular tyrosinase that were augmented by α-MSH stimulation (Figure 2A), resulting from the diminished content of tyrosinase proteins in the cells (Figure 3B). It is assumed that reduced contents of cellular tyrosinase are associated with accelerated proteolysis of tyrosinase via a proteasomal pathway [5]. To confirm this, we evaluated the effects of CMFE on the changes in the protein levels of tyrosinase in the presence or absence of the proteasomal inhibitor MG132. As shown in Figure 4, tyrosinase protein levels were increased by treatment with α-MSH and MG132. In addition, tyrosinase protein levels reduced by CMFE were increased upon MG132 treatment. These results show that CMFE is involved in proteasome degradation of tyrosinase, reducing intracellular tyrosinase protein levels.

### 2.5. Effects of CMFE on p38 Activation

Activation of p38 MAPK signaling accelerates the breakdown of melanogenic enzymes, including tyrosinase, to prevent excessive production of melanin [15]. To determine the relationship between CMFE-induced tyrosinase proteasomal degradation and the activation of p38, we investigated the effects of CMFE on the activation of p38 in the presence or absence of α-MSH stimulation. In α-MSH-untreated B16F10 cells, p38 was gradually activated according to the CMFE treatment time (0–6 h) (Figure 5A,B). Also, α-MSH activated p38 in B16F10 cells, and the activation of p38 by α-MSH was further increased by CMFE (Figure 5C,D). These results show that CMFE is involved in the proteasomal degradation of tyrosinase through the promotion of p38 activation.

## 3. Discussion

*C. monnieri* encompasses a diverse array of compounds, including osthol and several coumarin derivatives, such as imperatorin, xanthotoxin, isopimpinellin, and bergapten [14,16]. Traditionally, *C. monnieri* fructus has been utilized for addressing a spectrum of health issues, including skin-related conditions, due to its reputed antipruritic, anti-allergic, antidermatophytic, antibacterial, antifungal, and anti-osteoporotic properties [14,17,18]. However, despite its potential, there remains a paucity of research investigating its skin-whitening capabilities, particularly its anti-melanogenic effects.

The quest for lighter skin has gained momentum globally, driven by cultural and cosmetic preferences aimed at reversing sun damage and achieving a more youthful appearance. While hydroquinone has long been the go-to skin-lightening agent, its use in cosmetics has been restricted in Europe due to carcinogenic concerns since 2001. Consequently, there is an urgent need for safer and more effective alternatives.

Considering these considerations, our study sought to explore the anti-melanogenic potential of CMFE in response to α-MSH-induced melanogenesis in B16F10 cells. Remarkably, we observed that CMFE, particularly at a concentration of 5 μg/mL, effectively suppressed melanin production without inducing significant cytotoxicity (Figure 1).

The process of melanin synthesis is chiefly mediated by the enzyme tyrosinase, prompting extensive investigations into tyrosinase modulators. While various competitive or non-competitive inhibitors can curtail tyrosinase activity, our findings indicate that the suppressive effect of CMFE on melanin production does not stem from direct inhibition of tyrosinase activity, as evidenced by its ineffectiveness against mushroom tyrosinase (Figure 2B). Instead, CMFE notably reduced the tyrosinase activity in *α*-MSH-stimulated B16F10 cells (Figure 2A), suggesting a distinct mechanism of action.

Prior research has underscored the pivotal role of MITF in transcriptionally regulating melanogenic enzymes, including tyrosinase, TRP-1, and TRP-2, thereby modulating melanogenesis in melanoma cells [19,20,21,22,23]. Surprisingly, our study revealed a paradoxical upregulation of MITF and tyrosinase mRNA expression levels following CMFE treatment, even in the presence of α-MSH. However, the protein levels of MITF and tyrosinase were significantly diminished (Figure 3), implicating post-transcriptional regulatory mechanisms.

Given the intricate control of tyrosinase protein levels through synthesis and degradation, we investigated the involvement of proteasomal degradation pathways. Our results demonstrate that the CMFE-induced reduction in tyrosinase activity correlates with a decline in intracellular tyrosinase protein levels that is mediated by proteasome degradation (Figure 4).

Furthermore, the MAP kinase family member p38 has emerged as a key regulator of melanogenesis, with studies elucidating its role in MITF phosphorylation and subsequent ubiquitin-dependent proteasome degradation [24,25]. Accordingly, our investigation revealed that CMFE promotes the phosphorylation of p38, thereby facilitating the proteasomal degradation of tyrosinase (Figure 5).

Notably, osthol, a major compound in CMFE, has been documented for its multifaceted pharmacological activities, including anti-fibrotic, anti-tumor, anti-inflammatory, neuroprotective, and melanogenesis-inhibitory effects [26,27,28,29,30,31].

In summary, our findings shed light on the potential of CMFE as a natural whitening agent by elucidating its role in inhibiting melanin production through the proteasome-mediated degradation of tyrosinase via p38 activation. The elucidation of these mechanisms underscores the promising applications of CMFE in cosmetic science and warrants further exploration of its therapeutic potential in clinical settings. Moreover, our study contributes to the broader understanding of natural compounds as alternative treatments for skin pigmentation disorders.

## 4. Materials and Methods

### 4.1. Materials

The CMFE sample used in this study was provided by CNC Korea (Seoul, Korea) and was dissolved in dimethyl sulfoxide (DMSO; Sigma-Aldrich, St. Louis, MO, USA). Arbutin, mushroom tyrosinase, α-melanocyte-stimulating hormone (α-MSH), L-ascorbic acid, L-DOPA, methanol, MG132, and phenylmethylsulfonyl fluoride (PMSF) were purchased from Sigma-Aldrich. Dulbecco’s modified Eagle’s medium (DMEM) and fetal bovine serum (FBS) were purchased from Gibco (Grand Island, NY, USA). Phosphate-buffered saline (PBS) was purchased from Biosesang (Seongnam, Korea). Primary antibodies against MITF (microphthalmia-associated transcription factor), tyrosinase, and β-actin were purchased from Santa Cruz Biotechnology (Santa Cruz, CA, USA). p38 and phospho-p38 (Thr180/Tyr182) were purchased from Cell Signaling (Beverly, MA, USA). HPLC-grade MeOH and water were purchased from Burdick & Jackson (Muskegon, MI, USA). Unless otherwise indicated, all other chemicals were purchased from Sigma-Aldrich.

### 4.2. Plants and Materials

*C. monnieri* fructus was provided by CNC Korea (Seoul, Korea), and its voucher specimen (KHU-BMRI-201804) was deposited at the Bio-Medical Research Institute, Kyung Hee University, Yongin, Korea. The dried *C. monnieri* fructus (10 g) was extracted in 30% EtOH (200 mL × 2) at room temperature for 2 h, filtered, and evaporated under reduced pressure (CMFE). The CMFE used for in vitro experiments was diluted to a concentration of 10 mg/mL and stored in a −80 °C freezer. An ODS semi-prep LC system (Waters 600S with a Waters 2487 UV detector, Milford, MA, USA) with a semi-preparative C18 column (Supelcosil LC-18 Semi-prep, 5 μm, 250 × 10 mm, Supelco, Bellefonte, PA, USA) was used for the isolation of osthol. The obtained CMFE (300 mg) was subjected to ODS semi-preparative c.c. with the isocratic mobile phase consisting of MeOH:water (3:1) for 30 min at a flow rate of 2 mL/min under a 254 nm UV wavelength (Appendix A). A total of 35.6 mg of osthol was separated at 23.6 min and its NMR spectra (^1^H- and ^13^C) were recorded on a Bruker Avance 600 (Billerica, MA, USA) (Appendix A).

Osthol: ^1^H-NMR (600 MHz, CDCl_3_, *δ*_H_) 6.25 (d, 9.0, H-3), 7.63 (d, 9.0, H-4), 7.31 (d, 8.4, H-5), 6.85 (d, 8.4, H-6), 3.55 (d, 6.6, H-1′), 5.25 (d, 6.6, H-2′), 1.86 (s, H-4′), 1.69 (s, H-5′), 3.94 (s, H-7-OCH_3_); ^13^C-NMR (150 MHz, CDCl_3_, *δ*_C_) 161.4 (C-2), 112.8 (C-3), 143.7 (C-4), 113.0 (C-4a), 126.2 (C-5), 107.3 (C-6), 160.2 (C-7), 118.0 (C-8), 152.8 (C-8a), 21.9 (C-1′), 121.1 (C-2′), 132.7 (C-3′), 25.8 (C-4′), 17.9 (C-5′), 56.1 (C-7-OCH_3_).

### 4.3. Cell Culture

B16F10 mouse melanoma cells were purchased from the Korean Cell Line Bank (KCLB; Seoul, Korea) and cultured using DMEM medium (Hyclone, Logan, UT, USA) containing 10% FBS and 1% penicillin/streptomycin (Life Technologies, Grand Island, NY, USA) in a humidified atmosphere of 5% CO_2_ at 37 °C.

### 4.4. Cell Viability

Cell viability was measured using quantitative colorimetric WST-1 assay (EZ-Cytox, Dogen, Korea). The sodium salt of 4-[3-(4-iodophenyl)-2-(4-nitrophenyl)-2H-5-tetrazolio]-1,3-benzene disulfonate (WST-1) produces a highly water-soluble formazan through mitochondrial dehydrogenases. B16F10 cells were seeded at 2 × 10^3^ cells/well in 96-well tissue culture plates and treated with different doses (0–20 µg/mL) of CMFE for 48 h. After incubation with 10% WST-1 solution for 30 min, the optical density of formazan in the cells was measured at 450 nm using a microplate reader (SpectraMax^®^i3x Multi-Mode Detection Platform; Molecular Devices, Sunnyvale, CA, USA). Wells treated with 0 μg/mL served as the control. The cell viability of the vehicle control was established as 100%, comparing cell viability between the experimental and control groups.

### 4.5. Measurement of Melanin Content

B16F10 cells were seeded on 60 mm culture dishes at a cell density of 2 × 10^5^ cells per dish. The next day, the B16F10 cells were treated with α-MSH (100 nM) in the presence or absence of different doses of CMFE (0.5, 1, and 5 μg/mL) for 48 h. Arbutin (1 mM) was used as a positive control. The cells were washed with D-PBS and lysed with lysis buffer containing 50 mM sodium phosphate buffer (pH: 6.8), 1% Triton X-100, and 0.1 mM PMSF. After collecting the cell supernatant by centrifugation, the pellets were dissolved in 1 N NaOH for 1 h at 60 °C. The absorbance was measured at 490 nm with a microplate reader. Cell supernatants were used for protein quantification. The quantity of protein in each sample was measured using a BCA protein assay kit (Thermo Fisher Scientific, Waltham, MA, USA). The cellular melanin content was normalized to total protein concentration and was expressed as a percentage of untreated control cells.

### 4.6. Intracellular Tyrosinase Activity Assay

B16F10 cells were treated with α-MSH (100 nM) in the presence or absence of different doses of CMFE (0.5, 1, and 5 µg/mL) for 48 h. The negative control group used α-MSH-untreated B16F10 cells. The culture medium was then removed, and the cells were washed with D-PBS and lysed with lysis buffer containing 50 mM sodium phosphate (pH: 6.8) buffer, 1% Triton X-100, and 0.1 mM PMSF. After centrifugation, tyrosinase activity was determined in the cell supernatant by the addition of reaction mixture (40 µL of 100 mM sodium phosphate buffer (pH: 6.8) and 160 µL of 10 mM L-DOPA) in the presence of cell lysate (40 µg) for 1 h. The absorbance was measured at 490 nm with a microplate reader.

### 4.7. Mushroom Tyrosinase Inhibition Assay

The mushroom tyrosinase inhibitory activity of CMFE was determined by a colorimetric method [9]. A total of 40 µL of mushroom tyrosinase (1000 units/mL) was added to 100 µL of reaction mixture containing 0.1 M potassium phosphate buffer (pH 6.5), and 40 µL of L-DOPA (10 mM) was added in the presence or absence of CMFE. L-ascorbic acid (20 μg/mL) was used as a positive control for tyrosinase inhibition. The reaction was conducted at 37 °C for 10 min, and the absorbance was measured at 490 nm. Tyrosinase activity was expressed as a percentage of the control group.

### 4.8. Quantitative RT-PCR Analysis

B16F10 cells were treated with α-MSH (100 nM) in the presence or absence of different doses of CMFE (0.5, 1, and 5 µg/mL) for 24 h. Total RNA was extracted using TRIzol reagent (Invitrogen, Carlsbad, CA, USA), and cDNAs were synthesized using a Moloney Murine Leukemia Virus Reverse Transcriptase (Invitrogen). HOT FIREPol EvaGreen qPCR Mix Plus (ROX) (Solis BioDyne, Tartu, Estonia) was used for PCR, and cDNA was amplified using specific MITF (forward, 5′-GGAACAGCAACGAGCTAAGG-3′; reverse, 5′-TGATGATCCGATTCACCAGA-3′), tyrosinase (forward, 5′-CAAGTACAGGGATCGGCCAAC-3′; reverse, 5′-GGTGCATTGGCTTCTGGGTAA-3′), and *β*-actin (forward, 5′-CCCTGTATGCCTCTGGTC-3′; reverse, 5′-GTCTTTACGGATGTCAACG-3′) primers. Quantitative real-time PCR reactions were performed on a StepOne Plus Real-Time PCR System (Applied Biosystems, Foster City, CA, USA). Relative quantitative evaluation of each gene was performed using the comparative cycle threshold method [14].

### 4.9. Western Blot Analysis

For measuring the protein levels of MITF and tyrosinase, B16F10 cells were treated with α-MSH (100 nM) in the presence or absence of CMFE (5 µg/mL) for 24 h. Different cell lysates were obtained from the same experimental setting, followed by MG132 (10 μM) treatment for 6 h to test the effects of proteasomal inhibition on the expression levels of tyrosinase. Cells were lysed with PRO-PREP™ Protein Extraction Buffer (Intron Biotechnology, Seongnam, Korea) for 1 h on ice. After centrifugation at 13,000× *g* for 30 min at 4 °C, lysates were collected, and the protein concentrations were determined using a BCA protein assay. Equal amounts of protein were subjected to 12% SDS-polyacrylamide gel electrophoresis and electrophoretically transferred to polyvinylidene difluoride (PVDF) membranes (Millipore, Billerica, MA, USA). The membranes were blocked with 5% skim milk and incubated with primary antibodies against MITF, tyrosinase, p38, and phospho-p38 (Thr180/Tyr182) at a dilution factor of 1:2000. The blots were developed using horseradish peroxidase (HRP)-conjugated secondary antibodies (1:3000). Blots were re-probed with anti-β-actin antibody to verify equal protein loading. Protein bands were detected with Clarity Western ECL Substrate (BioRad, Hercules, CA, USA) according to the manufacturer’s instructions. The intensities of the bands were analyzed using a ChemiDoc Touch Imaging System (Bio-Rad, Hercules, CA, USA).

### 4.10. Statistical Analysis

Results are presented as the mean ± SEM (standard error of the mean) of three independent experiments. Data were analyzed by one-way analysis of variance (ANOVA) followed by Tukey’s test (GraphPad Prism 5 Software; San Diego, CA, USA). Values of *p* < 0.05 were considered to indicate statistical significance.

## 5. Conclusions

In conclusion, our study demonstrates that CMFE (especially at 5 μg/mL) effectively inhibits α-MSH-induced melanogenesis in B16F10 cells. The inhibitory mechanism of CMFE on melanogenesis appears to be associated with the promotion of tyrosinase degradation rather than the direct inhibition of tyrosinase protein expression. Furthermore, we identified the activation of p38 as one of the underlying mechanisms involved in tyrosinase degradation through the proteasomal pathway mediated by CMFE. These findings suggest that CMFE has the potential to serve as a natural whitening agent by inhibiting melanin production through the promotion of the proteasome-mediated degradation of tyrosinase via p38 activation. The elucidation of these mechanisms provides valuable insights into the development of novel therapeutic strategies for skin pigmentation disorders and highlights the promising applications of CMFE in cosmetic science.

## Figures and Tables

**Figure 1 plants-13-01305-f001:**
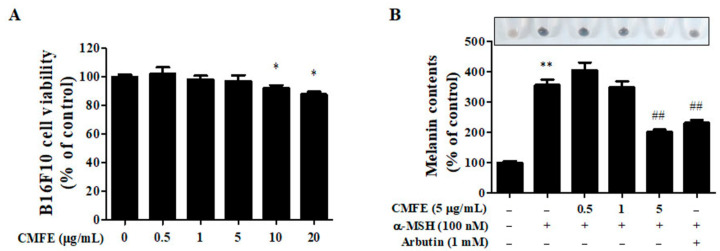
Cell cytotoxicity of CMFE and its effects on melanin production. (**A**) B16F10 cells were treated with CMFE at concentrations of 0.5–20 μg/mL for 48 h. Cell viability was determined using WST-1 assay. (**B**) B16F10 cells were treated with α-MSH (100 nM), arbutin (1 mM), and CMFE (0.5–5 μg/mL) for 48 h. Melanin content was determined by measuring the absorbance at 490 nm. Data are means ± SEM (n = 3) and are expressed as percentages of the untreated control. * *p* < 0.05, ** *p* < 0.01 vs. untreated control. ^##^ *p* < 0.01 compared with cells treated with α-MSH only. CMFE = C. monnieri fructus extract.

**Figure 2 plants-13-01305-f002:**
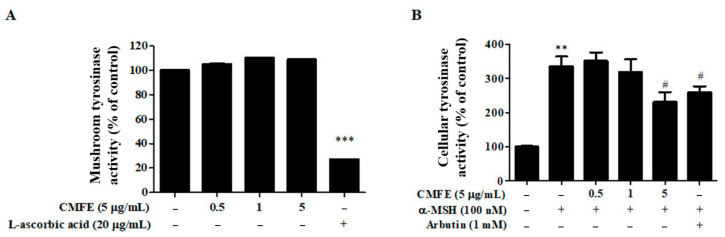
Effects of CMFE on the activities of cellular and mushroom tyrosinase. (**A**) B16F10 cells were treated with α-MSH in the presence or absence of CMFE (0.5–5 μg/mL) for 48 h. Cellular tyrosinase activity was measured using 3,4-dihydroxyphenylalanine (L-DOPA) as the substrate. (**B**) Mushroom tyrosinases were reacted with CMFE, and their respective activities were determined using L-DOPA as the substrate in a cell-free system. Data are mean ± SEM (n = 3) and are expressed as a percentage of the untreated control. ** *p* < 0.01, *** *p* < 0.001 vs. untreated control. ^#^ *p* < 0.05 compared with cells treated with α-MSH only. CMFE = C. monnieri fructus extract.

**Figure 3 plants-13-01305-f003:**
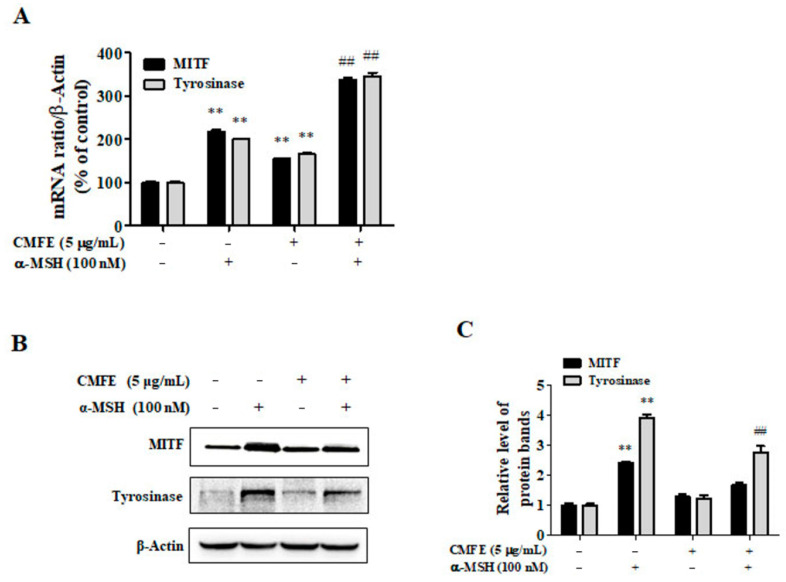
Effects of CMFE on MITF and tyrosinase expression levels. B16F10 cells were treated with 5 μg/mL CMFE in the presence or absence of α-MSH (100 nM) for 24 h. (**A**) RT-qPCR determined the mRNA expression levels of MITF and tyrosinase. (**B**) The protein expression levels of MITF and tyrosinase were determined by Western blot. (**C**) Relative levels of MITF and tyrosinase protein were compared to the untreated control. Data are mean ± SEM (n = 3) and are expressed as a percentage of the untreated control. ** *p* < 0.01 vs. untreated control. ^##^ *p* < 0.01 compared with cells treated with α-MSH only. CMFE = *C. monnieri* fructus extract.

**Figure 4 plants-13-01305-f004:**
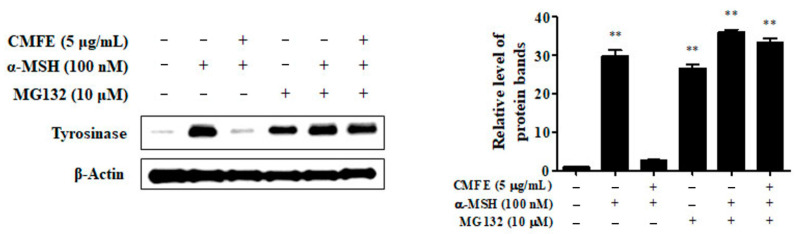
Effects of CMFE on proteasomal inhibition of tyrosinase protein. B16F10 cells were treated with α-MSH and 5 μg/mL CMFE for 24 h. Then, a membrane-permeable proteasome inhibitor MG132 (10 μM) was applied for 6 h. Cell lysates with or without MG132 treatment were subjected to Western blot. ** *p* < 0.01 vs. untreated control. CMFE = *C. monnieri* fructus extract.

**Figure 5 plants-13-01305-f005:**
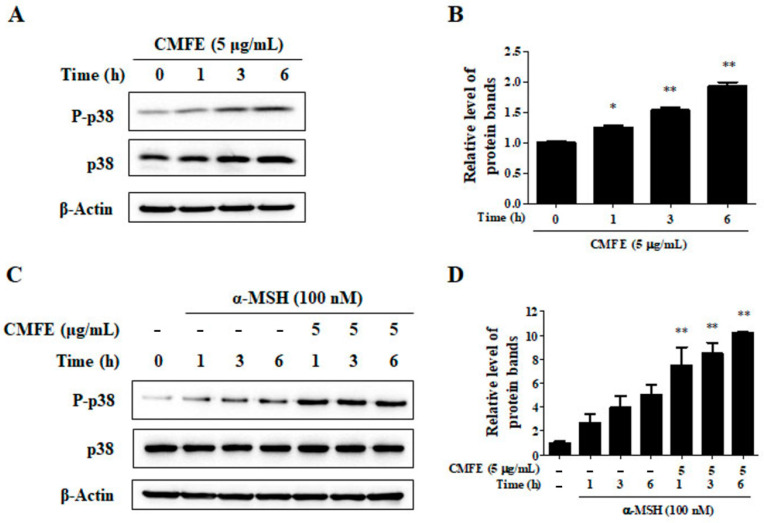
Effects of CMFE on p38 activation. B16F10 cells were treated with 5 μg/mL CMFE for 0–6 h in the absence (**A**,**B**) or presence (**C**,**D**) of α-MSH (100 nM). Cell lysates were subjected to Western blot for detection of p38 and its phosphorylated form (p-p38) (**B**,**D**). * *p* < 0.05, ** *p* < 0.01 vs. untreated control. CMFE = *C. monnieri* fructus extract.

## Data Availability

The data presented in this study are available from the corresponding author on request.

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
