# Peer review of "Anti-Melanogenic Effects of *Cnidium monnieri* Extract via p38 Signaling-Mediated Proteasomal Degradation of Tyrosinase"

_plants, 2024, doi:10.3390/plants13101305_

Round 1

Reviewer 1 Report

Comments and Suggestions for Authors

Attach file

Comments on the Quality of English Language

No comments

Author Response

The manuscript ID plants-2983743 is entitled: “Anti-melanogenic effects of Cnidium monnieri extract via p38 signaling-mediated proteasomal degradation of tyrosinase”.

The authors described the investigation the anti-melanogenesis effect and mechanism of action of Cnidium monnieri fructus extract (CMFE) in α-MSH-stimulated B16F10 cells. The study report that treatment with CMFE up to 5 μg/mL showed no significant cytotoxicity. CMFE significantly reduced α-MSH-stimulated melanin production (43.29 ± 3.55% decrease and cellular tyrosinase activity (31.14 ± 3.15% decrease.

Figures are in good quality and with adequate information for justified the biological assay.

References are important to justified the results obtained.

In conclusion: These findings suggest that CMFE has the potential to be a natural whitening agent for inhibiting melanogenesis. CMFE can be used as a natural whitening agent to inhibit melanin production.

The manuscript is suitable for publication in Plants

  • Thank you very much for your valuable suggestions. Based on your feedback, We have revised the overall content accordingly.

Minor corrections: p.2 line 54-55: …hydroxylation of L-tyrosine to L-3,4-dihydroxyphenylalanine (L-DOPA) and oxidation of L-DOPA to L-dopaquinone. … change by … hydroxylation of L-tyrosine to L-3,4-dihydroxyphenylalanine (L-DOPA) and oxidation of L-DOPA to L-dopaquinone. …. Check throughout the manuscript

  • Thank you for the feedback. We have revised the sentence as you suggested.

Reviewer 2 Report

Comments and Suggestions for Authors

This manuscript reports the anti-melanogenesis effect and mechanism of action of Cnidium monnieri. fructus extract in α-MSH-stimulated B16F10 cells via p38 signaling-mediated proteasomal degradation of tyrosinase. C. monnieri is a species widely studied due to its medicinal potential and use in traditional Oriental medicine, with dermatological importance. Its pharmacological use is mainly associated with osthol and other coumarin compounds. Scientific investigations with focus on the evaluation of the anti-melanogenesis effect of medicinal plants and their mechanism of action are welcome because they generally assume an important character related to the discovery of new drugs. The aims of the manuscript are interesting, and the experimental steps were apparently well conducted. However, there are some points that need explanation and/or correction. Please see the comments below.

1- Abstract:

- Line 18: “Cnidium monnieri (L.) fructus is widely...” – please rephrase to: “Cnidium monnieri fructus is widely...”.

2- Keywords: please do not use words that already appear in the title.

3- Introduction:

- Please update the references used in this section and provide more recent information on the topic of the manuscript.

- Line 62: please change “Umbelliferae family” to “Apiaceae family”.

- Line 80: “in vitro” – provide the italic form.

4- Results:

- Please see the captions of all figures and rephrase “CMFE; C. monnieri fructus extract” to “CMFE= C. monnieri fructus extract”.

- Line 117: “...3.15% inhibition, p<0.05) (Figure 2A).” – wouldn't that be Figure 2B?

- Line 122: “... As shown in Figure 2B, CMFE didn’t inhibit the activity of mushroom...” – this information corresponds to Figure 2A – please review.

- Figure 2A: please review the y-axis title.

- Figure 3C is not mentioned in the text.

- Line 182: “... by CMFE (Figure 5B-D).” – wouldn’t that be Figures 5C and D?

- Line 186: “...or presence (B) of α-MSH (100 nM)” – wouldn’t that be Figure C?

- Line 187: “...and its phosphorylated form (p-p38) (C-D)” – wouldn’t that be Figures B and D? Please review Figure 5 – the order of presentation is inconsistent with what is mentioned in the text.

5- Discussion

This section must be improved. The discussion should be more objective and provide more details on recent studies on the anti-melanogenic effects addressed. The authors should clarify this issue in order to highlight and justify the importance of this study. Please use more current references.

 - Please see also:

- Line 190: “C. monnieri (L.) Cuss. contains more than 350 compounds...” – please rephrase to: “C. monnieri contains more than 350 compounds...”;

- Line 192: “Traditionally, fructus C. monnieri ...” – please rephrase to: “Traditionally, C. monnieri fructus ...”.

6- Material and Methods

- Line 253: “C. Monnoeri fructus was provided by...” – please rephrase to “C. monnieri fructus was provided by...".

- Line 255: “Kyung Hee university...” – please rephrase to: “Kyung Hee University...”

- Line 255: “The dried C. Monnoeri fructus...” - “The dried C. monnieri fructus...”

- Lines 257-259: this sentence is confusing – please rephrase.

- Lines 259-260: please transfer this information to item 4.1 Materials.

- Lines 265-269: these data are unrelated to the text – please review.

- Lines 277-285: please provide controls for evaluating cell viability.

- Line 281: please provide at what concentration of DMSO the CMFE was dissolved.

- Lines 286-297: please provide negative control.

- Lines 289, 300 and 317: please provide at what concentration of DMSO the CMFE was dissolved.

- Line 311: please provide the concentration of CMFE used and in which concentration of DMSO the CMFE was dissolved.

7- Conclusions

The conclusion presents a summary of the results. It should finalize the findings presented and point out perspectives for the advancement of knowledge in the area studied – please review.

 In my final comments, I recommend that the manuscript should be reviewed by the authors. Introduction, Material and methods, Results, Discussion and Conclusions sections must be rephrased to explain more concisely the correlation between the variables studied.

Moderate editing of English language is required.

Comments on the Quality of English Language

Moderate editing of English language is required.

Author Response

This manuscript reports the anti-melanogenesis effect and mechanism of action of Cnidium monnieri. fructus extract in α-MSH-stimulated B16F10 cells via p38 signaling-mediated proteasomal degradation of tyrosinase. C. monnieri is a species widely studied due to its medicinal potential and use in traditional Oriental medicine, with dermatological importance. Its pharmacological use is mainly associated with osthol and other coumarin compounds. Scientific investigations with focus on the evaluation of the anti-melanogenesis effect of medicinal plants and their mechanism of action are welcome because they generally assume an important character related to the discovery of new drugs. The aims of the manuscript are interesting, and the experimental steps were apparently well conducted. However, there are some points that need explanation and/or correction. Please see the comments below.

1- Abstract:

- Line 18: “Cnidium monnieri (L.) fructus is widely...” – please rephrase to: “Cnidium monnieri fructus is widely...”.

  • Thank you. We have made the revisions as you suggested.

2- Keywords: please do not use words that already appear in the title.

  • Thank you. We have made the revisions as you suggested.

3- Introduction:

- Please update the references used in this section and provide more recent information on the topic of the manuscript.

- Line 62: please change “Umbelliferae family” to “Apiaceae family”.

- Line 80: “in vitro” – provide the italic form.

  • Thank you. We have extensively revised the introduction part as you suggested.

4- Results:

- Please see the captions of all figures and rephrase “CMFE; C. monnieri fructus extract” to “CMFE= C. monnieri fructus extract”.

- Line 117: “...3.15% inhibition, p<0.05) (Figure 2A).” – wouldn't that be Figure 2B?

- Line 122: “... As shown in Figure 2B, CMFE didn’t inhibit the activity of mushroom...” – this information corresponds to Figure 2A – please review.

- Figure 2A: please review the y-axis title.

- Figure 3C is not mentioned in the text.

- Line 182: “... by CMFE (Figure 5B-D).” – wouldn’t that be Figures 5C and D?

- Line 186: “...or presence (B) of α-MSH (100 nM)” – wouldn’t that be Figure C?

- Line 187: “...and its phosphorylated form (p-p38) (C-D)” – wouldn’t that be Figures B and D? Please review Figure 5 – the order of presentation is inconsistent with what is mentioned in the text.

  • Thank you. We have made the all revisions as you suggested.

5- Discussion

This section must be improved. The discussion should be more objective and provide more details on recent studies on the anti-melanogenic effects addressed. The authors should clarify this issue in order to highlight and justify the importance of this study. Please use more current references.

 - Please see also:

 - Line 190: “C. monnieri (L.) Cuss. contains more than 350 compounds...” – please rephrase to: “C. monnieri contains more than 350 compounds...”;

- Line 192: “Traditionally, fructus C. monnieri ...” – please rephrase to: “Traditionally, C. monnieri fructus ...”.

  • Thank you. We have extensively revised the Discussion part as you suggested.

6- Material and Methods

- Line 253: “C. Monnoeri fructus was provided by...” – please rephrase to “C. monnieri fructus was provided by...".

  • Thank you. We have made the revisions as you suggested.

- Line 255: “Kyung Hee Universit...” – please rephrase to: “Kyung Hee University...”

  • Thank you. We have made the revisions as you suggested.

- Line 255: “The dried C. Monnoeri fructus...” - “The dried C. monnieri fructus...”

  • Thank you. We have made the revisions as you suggested.

- Lines 257-259: this sentence is confusing – please rephrase.

  • Thank you. We have made the revisions as you suggested.

- Lines 259-260: please transfer this information to item 4.1 Materials.

  • Thank you. We have made the revisions as you suggested.

- Lines 265-269: these data are unrelated to the text – please review.

  • Thank you for your valuable indication. We've revised lines 265-269 to provide a more coherent description of the main components in CMFE. I hope this response is suitable for your suggestion.

- Lines 277-285: please provide controls for evaluating cell viability.

  • Thank you. As you indicated, we provided information on controls for evaluating cell viability in manuscript.
  • ‘The wells treated with 0 μg/mL served as the control. The cell viability of the vehicle control was established as 100%, comparing cell viability between the experimental and control groups.’

- Lines 286-297: please provide negative control.

  • Thank you. The negative control group used α-MSH-untreated B16F10 cells. And we added this information in manuscript.

- Line 281: please provide at what concentration of DMSO the CMFE was dissolved.

- Lines 289, 300 and 317: please provide at what concentration of DMSO the CMFE was dissolved.

- Line 311: please provide the concentration of CMFE used and in which concentration of DMSO the CMFE was dissolved.

  • Thank you. The stock sample was prepared by dissolving it in DMSO at a concentration of 10 mg/mL, and this information has been added to section 4.2.

7- Conclusions

The conclusion presents a summary of the results. It should finalize the findings presented and point out perspectives for the advancement of knowledge in the area studied – please review.

 In my final comments, I recommend that the manuscript should be reviewed by the authors. Introduction, Material and methods, Results, Discussion and Conclusions sections must be rephrased to explain more concisely the correlation between the variables studied.

Moderate editing of English language is required.

  • Thank you. As per your instructions, the conclusion has been extensively revised, and we have endeavored to rectify any grammatical errors to the best of our ability.

Reviewer 3 Report

Comments and Suggestions for Authors

The manuscript from Choi et al uses a variety of modern molecular biology techniques to demonstrate that a methanol extract of Cnidium monnieri has anti-melanogenesis activity due to its ability to enhance proteasomal degradation of tyrosinase. The experiments are well performed, clearly described, and carefully build upon each other to reach the conclusions reported. The supplementary material demonstrates that the extract contains two main compounds, one of which is identified as osthol. The manuscript mentions that osthol has been reported as having numerous activities, including melanogenesis-inhibitory activity, but it would be beneficial to demonstrate that the activities reported here are due to osthol, rather than another component of the extract. A footnote for the table included in the SI should state where the 400 MHz NMR data for Osthol comes from.

Comments on the Quality of English Language

The English is generally good, with only a few minor grammatical errors that editing should correct. The last two paragraphs of the introduction include many hyperbolic adjectives and could be toned down.

Author Response

The manuscript from Choi et al uses a variety of modern molecular biology techniques to demonstrate that a methanol extract of Cnidium monnieri has anti-melanogenesis activity due to its ability to enhance proteasomal degradation of tyrosinase. The experiments are well performed, clearly described, and carefully build upon each other to reach the conclusions reported. The supplementary material demonstrates that the extract contains two main compounds, one of which is identified as osthol. The manuscript mentions that osthol has been reported as having numerous activities, including melanogenesis-inhibitory activity, but it would be beneficial to demonstrate that the activities reported here are due to osthol, rather than another component of the extract. A footnote for the table included in the SI should state where the 400 MHz NMR data for Osthol comes from.

  • Thank you for your valuable suggestion. As you mentioned, we have now stated in the main text that osthol originates from CMFE and added references to the supplementary material.

The English is generally good, with only a few minor grammatical errors that editing should correct. The last two paragraphs of the introduction include many hyperbolic adjectives and could be toned down.

  • Thank you for your feedback. We carefully reviewed the last two paragraphs of the introduction to tone down the hyperbolic adjectives and improve the overall quality of the English. Additionally, we addressed minor grammatical errors through editing.

Round 2

Reviewer 2 Report

Comments and Suggestions for Authors

Dear authors,

Thank you for making the suggested corrections.

Congratulations for your paper.

Best regards!